# Industrial Soft Sensor Optimized by Improved PSO: A Deep Representation-Learning Approach

**DOI:** 10.3390/s22186887

**Published:** 2022-09-13

**Authors:** Alcemy Gabriel Vitor Severino, Jean Mário Moreira de Lima, Fábio Meneghetti Ugulino de Araújo

**Affiliations:** Computer Engineering and Automation Department, Federal University of Rio Grande do Norte, 3000 Senador Salgado Filho Avenue, Natal 59078-970, RN, Brazil

**Keywords:** particle swarm optimization, soft sensors, deep learning, stacked autoencoders, mutual information

## Abstract

Soft sensors based on deep learning approaches are growing in popularity due to their ability to extract high-level features from training, improving soft sensors’ performance. In the training process of such a deep model, the set of hyperparameters is critical to archive generalization and reliability. However, choosing the training hyperparameters is a complex task. Usually, a random approach defines the set of hyperparameters, which may not be adequate regarding the high number of sets and the soft sensing purposes. This work proposes the RB-PSOSAE, a Representation-Based Particle Swarm Optimization with a modified evaluation function to optimize the hyperparameter set of a Stacked AutoEncoder-based soft sensor. The evaluation function considers the mean square error (MSE) of validation and the representation of the features extracted through mutual information (MI) analysis in the pre-training step. By doing this, the RB-PSOSAE computes hyperparameters capable of supporting the training process to generate models with improved generalization and relevant hidden features. As a result, the proposed method can generate more than 16.4% improvement in RMSE compared to another standard PSO-based method and, in some cases, more than 50% improvement compared to traditional methods applied to the same real-world nonlinear industrial process. Thus, the results demonstrate better prediction performance than traditional and state-of-the-art methods.

## 1. Introduction

Numerous key-quality process variables are hard and high-cost to measure in real-time in complex industrial processes, specifically in the oil and chemical industries. In the lack of online measurements for such critical variables, efficient monitoring and control strategies may not be available. In such cases, inference approaches like soft sensors may surpass the above-cited problem [1]. Soft sensors can estimate the hard-to-measure variables using secondary variables, which are easy and low-cost to measure [2]. Several proposed methods have designed soft sensors, and models based on artificial intelligence techniques have succeeded in various applications, including industrial scenarios. Principal component regression (PCR), Support vector machine (SVM), Gaussian Process Regression (GPR), Partial Least Square (PLS), and Artificial Neural Network are among the most thriving and used methods [3,4,5,6,7,8,9,10,11,12,13].

Generally, in cases of building virtual sensors to measure quality variables in processes, there is not a lot of labeled data, but there is plenty of unlabeled data. In this case, semi-supervised methods are more promising alternatives than traditional methods, which demonstrate unsatisfactory performance when they have a limited amount of labeled data [14]. The extensive volume of unlabeled data stores latent information, which, when used correctly, can improve model reliability and prediction performance [13]. Deep learning strategies are increasingly being used in the implementation of semi-supervised methods [15]. A deep network architecture, known as a stacked autoencoder (SAE), which has its weights calculated by unsupervised pre-training and applied to the supervised fine-tuning step, is successfully used in many soft sensor designs applied in industrial processes [16,17,18,19,20].

However, one of the difficulties of deep neural networks applications such as SAE is the definition of your hyperparameters: batch size, learning rate, step hidden features, among others. Evaluating the many possible hyperparameter configurations would require a high cost of time. Thus, the task of defining hyperparameters is an optimization problem.

Exact algorithms are methods used to find an optimal solution to a given problem, in which no time limits are imposed on the search process. Consequently, exact algorithms require a high computational effort. An alternative to exact algorithms are meta-heuristics. These algorithms are inspired by nature, i.e., natural phenomena and/or physical laws. Meta-heuristics propose to combine basic heuristic methods to effectively explore a search space. For this reason, several meta-heuristics have been applied to the hyperparameters’ definition of deep neural networks, such as Bayesian Optimization (BO) [21,22], Genetic Algorithm (GA) [21,23], Harmony Search (HS) [24], Whale Optimization Algorithm (WOA) [25], and Particle Swarm Optimization (PSO) [26].

The work of [21] used four hyperparameter optimization methods to design a rotation angle estimator based on an artificial neural network: Random Search (RS), Hyperband (HB), BO, and GA. Among the methods, the BO and GA metaheuristics showed better results in selecting the hyperparameters in vast search space and strongly nonlinear problems. Ref. [22], meanwhile, optimized the hyperparameters and structure of a convolutional neural network (CNN) applied in a data-driven intelligent fault diagnosis technique for rotating machines. The results demonstrate the efficiency of BO in optimizing the hyperparameters and the network structure when the objective function is time, which is computationally expensive in the cases of CNNs. In [23], the authors proposed a strategy of hyperparameter optimization based on Non-dominated Sorting Genetic Algorithm-II (NSGA-II) in the training process of a deep learning model. The obtained results showed improved generalization error and lowered overfitting incidence ranting. Furthermore, the proposal [24] applies an HS metaheuristic to optimize the hyperparameters of a 1D CNN model, aiming for accuracy improvements in pattern recognition. The improved model has shown a better precision rate than the non-optimized CNN model. Ref. [25] presents the WOA metaheuristic for optimizing the hyperparameters of a neural network. The search behavior of humpback whales inspires the method. WOA found a good set of hyperparameters in a shorter period when compared to the GS method while being of similar quality to the more straightforward RS method. Ref. [26] proposes a novel soft sensing approach based on semi-supervised ensemble learning. A novel online model adaptation criterion approach accurately describes the relationships among samples and local models and can provide higher mensuration exactness. The parameters of the presented technique are completed automatically by the PSO method. The simulation outcomes reveal the significance of the suggested process in dealing with nonlinear regression problems.

The evaluation functions of the previously-mentioned methods only regard information based on error indices between the output generated by the models and the actual output of the systems as mean squared error. However, the evaluation functions do not consider the representation relevance of the features extracted in the pre-training stage, an essential metric for soft sensors based on deep learning. This work proposes the RB-PSOSAE, a Representation-Based Particle Swarm Optimization with a modified evaluation function to optimize the hyperparameter set of an SAE-based soft sensor. The evaluation function considers the mean square error (MSE), as in previous works, and the relevance of the information extracted from the unlabeled data set through mutual information (MI) analysis in the pre-training step. Doing so enhances PSO to compute hyperparameters capable of generating models with improved generalization and relevant hidden features, and then better performance than traditional-based PSO.

The main contributions of this research are as follows:Present an improved PSO for the automatic adjustment of hyperparameters of deep neural networks based on the relevance of extracted representations;Carry out the extraction of representative features through the analysis of mutual information used in the PSO evaluation function;Improve the performance of the SAE model used for feature extraction in the unsupervised learning stage;Obtain a neural model with relevant features generated from an optimal combination of hyperparameters using the unlabeled data.

The contributions mentioned above have been demonstrated to be acceptable and successful for the automatic adjustment of hyperparameters for an industrial plant of a debutanizer column. The work sections are as follows: Section 2 introduces theoretical fundaments. Section 3 explains the proposal in details. Furthermore, Section 4 presents the obtained results. Finally, Section 5 shows the conclusions and future perspectives of this work.

## 2. Preliminaries


### 2.1. Autoencoders

Autoencoder (AE) is an artificial neural network composed of two components: the encoder and the decoder [27]. The encoder aims to map its input into a low-dimensional, high-level, meaningful representation. Conversely, the decoder receives the encoder’s output and tries to rebuild the original input. In this process, the autoencoder can learn high-relevant features from input data in reconstructing the input by using extracted hidden features. Figure 1 illustrates an AE architecture [13].

The encoder receives input $x={\left[x_{1},x_{2},\cdots ,x_{n}\right]}^{T}\in R^{n}$ and transforms it into a low-dimensional hidden features $h={\left[h_{1},h_{2},\cdots ,h_{m}\right]}^{T}\in R^{m}$. Additionally, the decoder computes the previous-obtained hidden features to try to map out the input data. Equations (Equation 1) and (Equation 2) characterize the above-cited encoder and decoder operations:(1)$$h=f(W_{e}x+b_{e}),$$
(2)$$\hat{x}=g\left(W_{d}h+b_{d}\right),$$
where $W_{e}\in R^{n\times m}$, $b_{e}\in R^{m}$, $W_{d}\in R^{m\times n}$ and $b_{e}\in R^{n}$ are the weight matrices and bias of the encoder and decoder, respectively. Terms *f* and *g* are the commonly-used activation functions sigmoide or ReLU [28]. Figure 2 demonstrates the schematic diagram of an SAE.

The mean square error (MSE) among x and $\hat{x}$ represents the training loss function of AE as Equation (Equation 3) illustrates. The learning process adjusts the parameters set $\left(W_{e},b_{e},W_{d},b_{d}\right)$ to minimize the reconstruction error:(3)$$J_{AE}=\frac{1}{m}\sum_{i=1}^{m}\left(\frac{1}{2}{∥\hat{x_{i}}-x_{i}∥}^{2}\right)$$

When several AE are stacked, it builds a deep structure named stacked autoencoder (SAE), which is able to learn high-level features since each AE is an SAE layer. Basically, an SAE architecture uses the previous layer output as input to feed the next layer. Training such a model normally takes two stages: the unsupervised pre-training and the supervised fine-tuning. In the first stage, layer-by-layer pre-training tries to minimize the reconstruction loss function as Equation (Equation 3) illustrates. On the other hand, the second phase fine-tunes all SAE parameters by minimizing the prediction error [29,30].

An SAE model uses unlabeled and labeled data samples to develop semi-supervised soft sensors, but the model does not necessarily learn high-level representations for soft-sensing purposes. Naturally, unsupervised pre-training does not regard targeted-output data, which might degrade the prediction’s performance even when a successful fine-tuning performs [31]. In addition, an SAE model has several hyperparameters to be chosen, and each one of them can impact the model’s performance and extracted-features quality. Furthermore, the optimization of an SAE’s hyperparameters regarding the relevance of the extracted hidden features is critical to build a suitable SAE-based soft sensor. In this work, an enhanced particle swarm optimization, which considers features’ representation relevance, optimizes an SAE’s hyperparameters.

### 2.2. Particle Swarm Optimization

PSO was inspired by nature, based on the social characteristics of bands of birds and schools of fish in search of a nest or food [32]. PSO is widely applied in solving optimization problems [33] due to the characteristics of few parameters, simple formulas, easy implementation, and good convergence speed. Recent advances in the application of PSO are emerging in the optimization of hyperparameters of SAEs [34,35,36]. According to [37], the PSO is similar to the GA, as the system is initialized with a population of random candidate solutions, here called a swarm. However, this population presents differences because a velocity $v_{i}$, Equation (Equation 4), is assigned for each potential solution, and the potential solutions, called particles, are guided through the solution space. Each particle has a position $x_{i}$, Equation (Equation 5), within the search space. The best solution found by the *i*-th particle up to the iteration *k*, ${\mathrm{pbest}}_{i}$, is linked to this position. Another critical variable in the algorithm execution is the best global solution, gbest, which is the best solution found by the *i*-th swarm up to the iteration *k*. The velocity updating of each particle through pbest and gbest and, later, its position is the base of PSO operation: (4)$$v_{i}^{k+1}=\omega v_{i}^{k}+r_{1}\phi_{1}\left({\mathrm{pbest}}_{i}^{k}-x_{i}^{k}\right)+r_{2}\phi_{2}\left(\mathrm{gbest}-x_{i}^{k}\right),$$
(5)$$x_{i}^{k+1}=x_{i}^{k}+v_{i}^{k+1},$$
in which ω is the inertia factor, responsible for promoting a balance between global and local exploration. The inertia factor is related to the step of the method. If its value is too high, particles can pass through acceptable solutions without visiting them; on the other hand, if its value is too low, the particles may not sufficiently explore places that have acceptable solutions. The terms $\phi_{1}$ and $\phi_{2}$ represent the degree of confidence of the particle in the best solution found by it, ${\mathrm{pbest}}_{i}$, and in the best solution found by the swarm, gbest. If their values are too low, the particles will “fly” for more distant routes towards the target region, or if their values are too high, the particles will make abrupt movements towards the potential region. $r_{1}$ and $r_{2}$ are random numbers that range between 0 and 1.

## 3. The Proposed Method


This section details the proposal step-by-step. First, the proposed representation-based PSO performs hyperparameter optimization in the unsupervised pretraining of an SAE model, generating an optimized SAE named RBPSO-SAE. In the next stage, an LSTM structure couples with the RBPSO-SAE to proceed with the supervised fine-tuning.

### 3.1. Data Preprocessing

As the first step, the dataset {X,Y} is standardized into a range of [0,1]. Such preprocessing can improve the overall stability of the model. Unlabeled $\left\{X^{U}\right\}$ and labeled $\left\{X^{L},Y\right\}$ data compose the dataset with ratios of 90% data and 10%, respectively. This scenario emulates real industrial scenarios where labeled is scarce, but the number of unlabeled samples is abundant. The large set of non-labeled data can hide relevant features about the process that traditional techniques do not exploit. Then, the unsupervised pretraining uses the unlabeled data to train the RBPSO-SAE models, aiming extraction of meaningful features.

The training set ${\left\{X^{L},Y\right\}}_{Tr}$ represents 40% of the labeled set, and a total of 10% of the labeled set forms the validation set ${\left\{X^{L},Y\right\}}_{V}$, and, finally, the testing set ${\left\{X^{L},Y\right\}}_{Te}$ takes 50% from the labeled set. The supervised fine-tuning uses the three above-cited labeled subsets to train, validate, and test the entire deep architecture composed by the RBPSO-SAE coupled to an LSTM.

### 3.2. Representation-Based PSO

A regular AE aims to reconstruct the input data at its output, and in this process, it learns meaningful representations of the input data. In the learning process, a regular AE treats all available data similarly. However, it is not true that all variables are equally relevant to building AEs for soft sensing purposes. Once irrelevant information is on an AE-based soft sensor, it can damage prediction performance.

This work presents a representation-based PSO to optimize the hyperparameters for SAE’s pretraining. The proposed PSO evaluates two critical points: the test mean square error (MSE), which measures the capacity of reconstructing the inputs, and the mutual information (MI) between features and targeted outputs to analyze how relevant extracted features are.

Choosing hyperparameters is one of the most complex parts of training deep learning models due to their magnitude, variables’ volume, and underlying correlation. In addition, there is not a formal method to do it beyond empirical methods such as random and grid search, which does not guarantee the optimal hyperparameter set. Therefore, the representation-based PSO tries to balance MSE and MI to get an optimized hyperparameter set capable of generating an enhanced AE with relevant features and, then, better adapted for soft-sensing.

The following steps detail the proposal:

Step 1. MI analysis evaluates the nonlinear relationship between each extracted feature and target outputs. Suppose the calculated MI is not greater than an early-defined threshold value, representing the minimum required relevance. In that case, the computed feature $x_{i}$ is irrelevant to inferring the desired outputs as follows:(6)$$MI(x_{i},y)\leq th,$$
where th is the threshold value.

As the first step, the proposal determines the MI threshold value. A uniform distribution generates a 1000 random vectors with values in the range of [0,1]. MI analysis among each generated arbitrary vector and the targeted output is computed. After that, calculated MI values are ordered in a descending way, and the 50th value points to the th. As a result, when computed MI values are greater than th, the confidence level is 95%.

Step 2. The second step specifies the hyperparameters. The selected training hyperparameters are the batch size BS, the learning rate LR, and the number of hidden features HF. As an AE has an intrinsic characteristic to extract hidden features by reconstructing its input, reducing the number of hidden features by each layer boosts AE ability and representations as the model gets deep. Therefore, this proposal does not use a HF for each layer but one HF that decreases for each model layer proportionally.

Another concern is that the range of hyperparameters needs to be defined. There is no formal method to set up the hyperparameters to define their searching space. Determined inferior and superior range limits have to guarantee that PSO can use all relevant-available searching space.

Step 3. The third step defines the early-needed PSO parameters for running the optimization attempts: the number of swarms $N_{SW}$, the number of particles $N_{pr}$, the inertial factor ω, and confidence parameters $\Phi_{1}$ and $\Phi_{2}$. The $N_{SW}$ and $N_{pr}$ relate to the number of evaluated models. The higher these values, the greater the time spent and the computational cost. For this reason, choosing these values is a task where balance among the number of models and performance matters. The inertia factor ω relates to the search step of the optimization algorithms. A low inertia factor will result in a lower speed of the particles, that is, a smaller displacement. In comparison, a high value will cause a higher speed of the particles, consequently a more expressive displacement in the search space. Finally, $\Phi_{1}$ and $\Phi_{2}$ represent the particle’s confidence in itself and the best solution found by the group, respectively. These values relate to the degree of exploration of the search space, influencing particles’ convergence speed to find the best solution.

Step 4. Step four explains the representation-based PSO as follows:

Step 4.1. Once the PSO parameters are defined, we randomly generate a set of initial hyperparameters, the swarm $S_{0}$ with $P_{S_{0}}=\left\{P_{0},P_{1},...P_{N_{pr}}\right\}$ particles set regarding hyperparameters’ ranges. Each particle of $P_{S_{0}}$ represents a set with values of BS, LR, and HF that set up the hyperparameters for the SAE’s pretraining.

Step 4.2. By applying the first particle of $P_{S_{0}}$ set, the pretraining of the first stacked autoencoder gets started. The process applies all available particles in $P_{S_{0}}$ until the training process of $N_{pr}$-th model ends. In sequence, the proposal evaluates the set of representation-based PSO stacked autoencoders (*RBPSO-SAE*).

Step 4.3. This work evaluates each of the obtained *RBPSO-SAE* models regarding their ability to extract features and how relevant those features are for the main goal of the model, which is virtual sensing tasks. First, the proposal calculates the MSE between real and targeted outputs from the test set, evaluating the model’s performance. Second, MI analysis among target outputs and the model’s output performs the representation-based evaluation. All of the RBPSO-SAE models receive the labeled dataset $\left\{X^{L},Y\right\}$ as input, then they generate a representation-based output Φ, which are the high-level extracted features from the input data. The MI analysis of the nonlinear relationship among features of Φ and the targeted-output values *Y* regarding Equation (Equation 6). The higher the MI value is, the greater the relevance of the analyzed feature to estimate desired outputs. By using MSE and the mean of MI values, $MI_{mean}$, the approach presents a fitness function (Figure 3) used to measure the appropriate level of each model as follows:(7)$$fitness=\alpha *MSE+\frac{1}{\beta *MI_{mean}}$$
where α and β are values to tune the importance level of MSE and MI values in obtaining an optimized model. The lower the value of the fitness function, the better the solution. The proposed method updates the best global solution gbest and the pbest, which is the best solution found by the *i*-th particle up to the iteration *k*.

The above-described representation-based PSO process repeats itself, generates a new swarm, and evaluates the new particle-generated models.

Step 5. When the representation-based PSO finishes, the best generated global model composes the soft sensor’s architecture. An LSTM structure couples to the best-found RBPSO-SAE to accomplish the regression task. An LSTM model has the intrinsic ability to handle time-series input and its dynamics. As an industrial process is highly nonlinear and dynamic, LSTM suits soft-sensing purposes well. The LSTM receives the highly in-depth and relevant features from the RBPSO-SAE and then estimates the targeted-output values.

## 4. Case Studies and Results


The RBPSO-SAE-based soft sensor evaluation proceeds through an industrial-based plant debutanizer column case study. For comparison, this work utilizes the following models:Deep learning-based methods: SAE with grid search (GS-SAE) and SAE with random search (RS-SAE);Deep learning-based method with PSO optimization: SAE with PSO tuning hyperparameters through MSE only (PSO-SAE);Proposed relevant representation-based PSO soft sensor model: RBPSO-SAE.

The root-mean-square error (RMSE) and coefficient of determination ($R^{2}$) are the chosen metrics for comparing the above-cited methods’ prediction efficiency:(8)$$RMSE=\sqrt{\frac{1}{N_{Ts}}\sum_{i=1}^{N_{Ts}}{\left({\hat{y}}_{i}-y_{i}\right)}^{2}},$$
(9)$$R^{2}=1-\frac{\sum_{i=1}^{N_{Ts}}{\left({\hat{y}}_{i}-y_{i}\right)}^{2}}{\sum_{i=1}^{N_{Ts}}{\left(y_{i}-{\bar{y}}_{i}\right)}^{2}}.$$

Equations (Equation 8) and (Equation 9) compute *RMSE* and $R^{2}$, respectively. The terms $y_{i}$ and ${\hat{y}}_{i}$ represent the real and estimated output, while the $\bar{y}$ represents the mean value of actual outputs. The $N_{Ts}$ represents the number of samples in the testing set. The RMSE indicates the error between targeted and estimated values. It is a metric usually used to evaluate soft sensor performance, and it quantifies the deviation between predicted and actual values in a squared error sense [38]. Therefore, RMSE quantifies reliability and the prediction performance [39]. Beyond that, predicting quality variables has inherent uncertainty. The standard deviation (SD) of RMSE is the adopted metric to measure the uncertainty range of the attained results over different runs [40].

The $R^{2}$ represents a correlation between predicted and actual outputs, in the form of a variance value over the desired outputs, where a high value represents better performance and higher reliability of the model [41].

### 4.1. Industrial Debutanizer Column Process

The debutanizer columns are regular devices that process desulfurization and naphtha cracking in oil and gas refineries [42]. Withdrawing propane (C5) and butane (C4) from naphtha steam is the primary goal of such an apparatus. Consequently, the lower the amount of butane, the better the quality of the naphtha end products. However, no physical sensors can measure the amount of butane in real-time. One solution is to use soft sensors to estimate the butane concentration simultaneously. Figure 4 illustrates the debutanizer column and its devices. The squares containing gray circles represent the hardware sensors that measure process variables such as flow, pressure, and temperature. The purpose of the debutanizer column used is to remove C3 and C4 from naphtha steam. Reducing the C4 concentration increases the quality of the final product located at the bottom of the debutanizer column. Gas chromatographs measure C4 concentration. However, due to a long measurement interval, they cannot provide real-time C4 concentrations for monitoring and control purposes. As explained earlier, soft sensors can estimate real-time measurements unavailable from physical sensors, such as C4 concentration. In [43], the dataset and more process details are available.

Table 1 lists the process variables present in the debutanizer column process. The study-case debutanizer offers 2384 data samples for each process variable, with a sampling time $T_{s}$ = 6 min. However, in a real distillation column scenario, not all of the data are labeled. Therefore, the proposal uses only a tiny part of the data samples as labeled in order to reproduce real scenarios where labeled data are scarce. The fine-tuning stage utilizes 240 samples, representing only 10% of the total data. Then, the non-labeled dataset has 2144 data samples, representing 90% of available data applied in the unsupervised pretraining.

As a regular dynamic system, the study-case debutanizer column outputs are a product of current inputs and past outputs. A feature engineering approach can handle the dynamicity of the process: past input and output values incorporate the current input. For the proposed model, the input blends to $X=[u\left(t\right),...,u\left(t-d_{x}\right),y\left(t-1\right),...,y\left(t-d_{y}\right)]$, where *u* and *y* are inputs and outputs, and $d_{x}$ and $d_{y}$ time-delay of inputs and outputs, respectively. This work employs $d_{x}=d_{y}=6$.

### 4.2. RBPSO-SAE

To get started with RBPSO-SAE strategy, the proposal set has a total of 30 swarms with 10 particles each, thus adding up to 300 model evaluations. It is important to note that this does not necessarily mean that this approach trained 300 distinct models since particles with the same sets of hyperparameters are present in all swarms. A value of ω is equal to 0.01. This value is related to the search step of the optimization algorithms, so, interestingly, its value is neither high nor low considering the magnitudes of the hyperparameters. Furthermore, all hyperparameters use the same ω. The values 1.2 and 2.4 were set for $\phi_{1}$ and $\phi_{2}$, respectively. The chosen set-values mean that the particle has twice as much confidence in the best solution found by the swarms as in the best solution found by itself. Finally, the values that the hyperparameters can take belong to the ranges [0.0005,0.0100], [10,500], and [2,15] for learning rate (LR), batch size (BS), and hidden features (HF), respectively. However, they were normalized to the range [0,1] so that only one value of ω is necessary to set. Since all the parameters are properly set, the proposed RBPSO-SAE approach starts to build SAE’s model with optimized hyperparameters based on the mean square error between estimated and actual outputs, and the MI-based relevance of extracted features.

Figure 5 illustrates the behavior of *g-best* particle, which means the best-obtained SAE model through the generated swarms, regarding the fitness criteria: mean square error and mean of MI values between extracted features and targeted output. By observing the graph of the evolution of the MSE, the mean, and the standard deviation of the MI, we can notice the following: There is a sharp drop in the MSE value between the first and the fifth swarm, stabilizing between the sixth and the sixteenth swarm. Finally, it decreases from the seventeenth swarm and returns to stability at the twenty-sixth. The mean MI value starts stable, but there is a noticeable increase between the third and fourth swarm. Soon after, it remains stable between the fifth and the sixteenth swarm and then acquires a rising behavior until the twenty-sixth swarm. MI’s mean and standard deviation values start stable, showing a decrease between the second and third swarms. Next, its value remains stable between the fourth and fifteenth swarm. Finally, it shows a falling behavior between the sixteenth and twenty-first swarm, and then remains stable. The increase in the mean MI value and the decrease in the MSE value and MI standard deviation along the swarm’s evolution imply improvements in the projected SAE performance. A lower MSE value of the validation implies a better generalization. As well as a higher value of the mean of the MI, we obtain more representative features. A smaller value of the MI standard deviation means minor variation in the quality of the features extracted from the data. Thus, the previous graph proves the efficiency of enhanced-proposed PSO application in adjusting hyperparameters considering its evaluation of MI.

Table 2 compares the prediction performance of the models obtained by fitting the hyperparameters found by the algorithms applying traditional search methods of RSSAE and GSSAE, applying the PSOSAE meta-heuristic, and applying the proposed search method RS-PSOSAE. The algorithms that use the traditional search methods, RSSAE and GSSAE, obtain the worst results compared to the other methods. However, when using search methods that consider past results to find new solutions, the algorithms that apply metaheuristics have better results than traditional search methods. RB-PSOSAE obtains an improved result compared to the PSOSAE method, which is explained due to PSOSAE considering only MSE information in its evaluation function. The better result of the RB-PSOSAE happens because its evaluation function considers information from MSE and MI. Thus, only relevant representations are present in its acquired knowledge, making it more suitable for soft sensor applications. In addition, the prediction performance of the other methods was tested with the same debutanizer column process employed in this paper. Looking at Table 2, which contains the qualitative comparison of the methods, RB-PSOSAE showed the best result. PSOSAE, GSSAE, and RSSAE are methods that only consider the MSE in their evaluations. However, RB-PSOSAE, besides considering the MSE, also evaluates the searched solutions through MI analysis, explaining its improved performance. Furthermore, RB-PSOSAE has the lowest standard deviation (SD) of RMSE, which points to its stability under uncertain conditions. RB-PSOSAE, when compared to PSOSAE, showed a 16.4% improvement in RMSE value. Meanwhile, compared to GSSAE and RSSAE, it showed an improvement in RMSE values of 54.7% and 49.4%, respectively.

Figure 6 demonstrates the prediction results from the test dataset using parity plots. Again, RB-PSOSAE showed higher accuracy, as expected, compared to the other methods. In addition, Figure 7 illustrates the relative prediction errors with boxplots of the four methods, RB-PSOSAE, PSOSAE, GSSAE, and RSSAE, in descending order of performance. The box edges indicate the 25th and 75th percentiles, while the red mark in the center represents the median value within each box. The larger the box’s width, the more dispersed the prediction errors. The upper and lower whiskers represent maximum and minimum values. The narrower box range of RB-PSOSAE indicates better prediction performance among the four compared methods mainly because RB-PSOSAE, through its evaluation function, can select the most relevant representations, extract nonlinear features, and handle process dynamics. Nonetheless, there are uncertainties to be considered in industrial processes [13,44], so Figure 7 shows some outlines represented individually by red dots. Inadequate initial parameters and discrepant values result in improper soft sensors with high relative prediction error. In contrast, acceptable soft sensors obtain stationary relative prediction errors for the same data set. Thus, stationary error values show the robustness of the model [13,45]. Therefore, the soft sensor designed by the RB-PSOSAE achieved better performance, reliability, and robustness.

## 5. Conclusions


An improved PSO based on representational learning has been proposed and tested for automatic tuning hyperparameters of deep neural networks. The RB-PSOSAE combines the search strategies of the PSO meta-heuristic, high-level feature extraction, and mines relevant representations in the SAE layers through MI analysis. In the PSOSAE method, particle swarms search for hyperparameter sets of an SAE using massive amounts of unlabeled data. However, unsupervised SAE modeling does not guarantee to learn relevant representations for soft-sensing purposes. Therefore, the RB-PSOSAE aims to highlight the most significant features, retain the relevant ones, and remove the irrelevant ones. The obtained results demonstrated that RB-PSOSAE improved prediction performance compared to traditional search methods and meta-heuristics that only consider information from the MSE and do not deal with the dynamics of the process. Furthermore, RB-PSOSAE was more reliable and robust, as demonstrated in the same case study and under the same conditions. However, despite the contributions presented, future work could analyze the performance of the proposed method in other case studies, besides investigating the implementation of different metaheuristics and improving the evaluation function used.

## Figures and Tables

**Figure 1 sensors-22-06887-f001:**
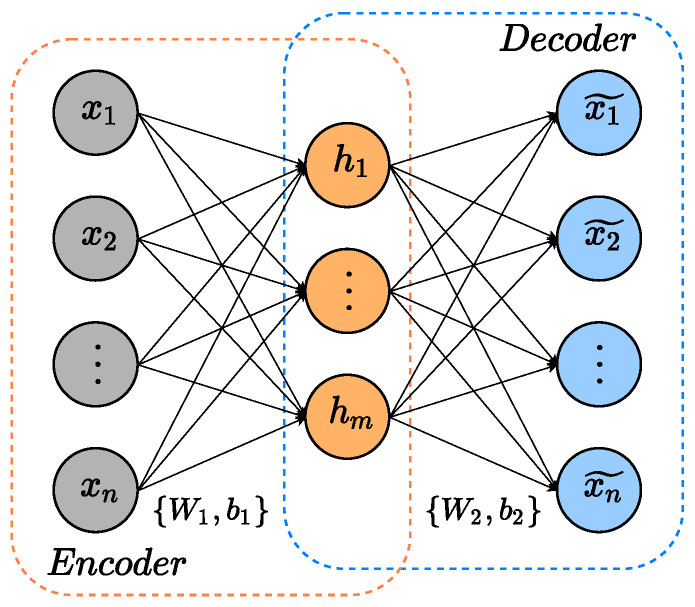
Basic AE schematic.

**Figure 2 sensors-22-06887-f002:**

Stacked Autoencoders schematic diagram.

**Figure 3 sensors-22-06887-f003:**
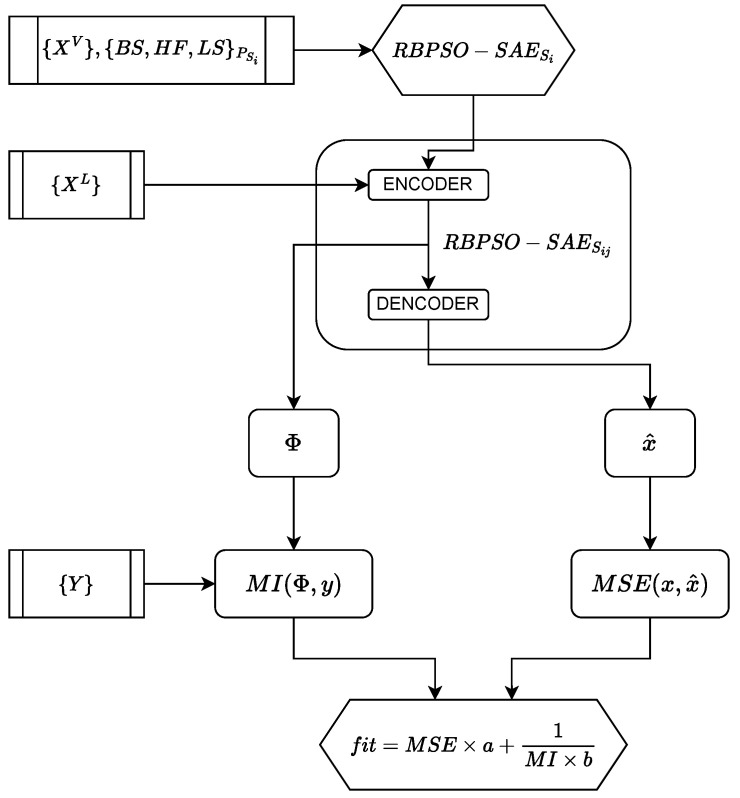
Fitness function flowchart.

**Figure 4 sensors-22-06887-f004:**
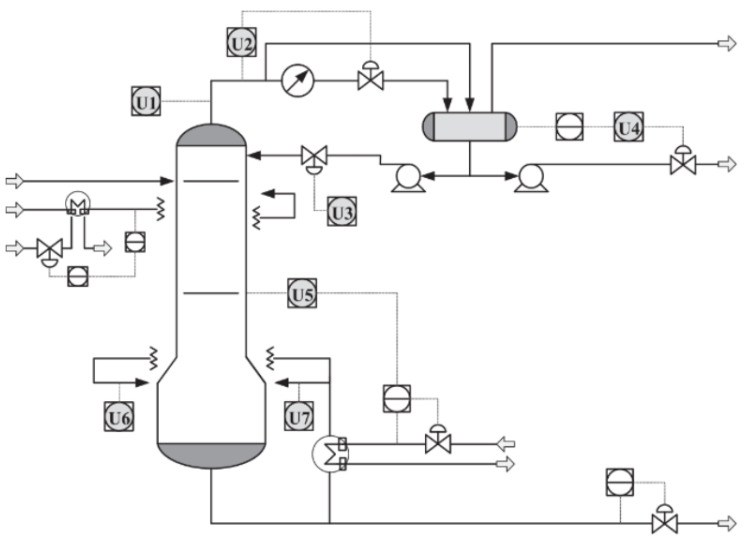
Schematic representation of the debutanizer column process [13].

**Figure 5 sensors-22-06887-f005:**
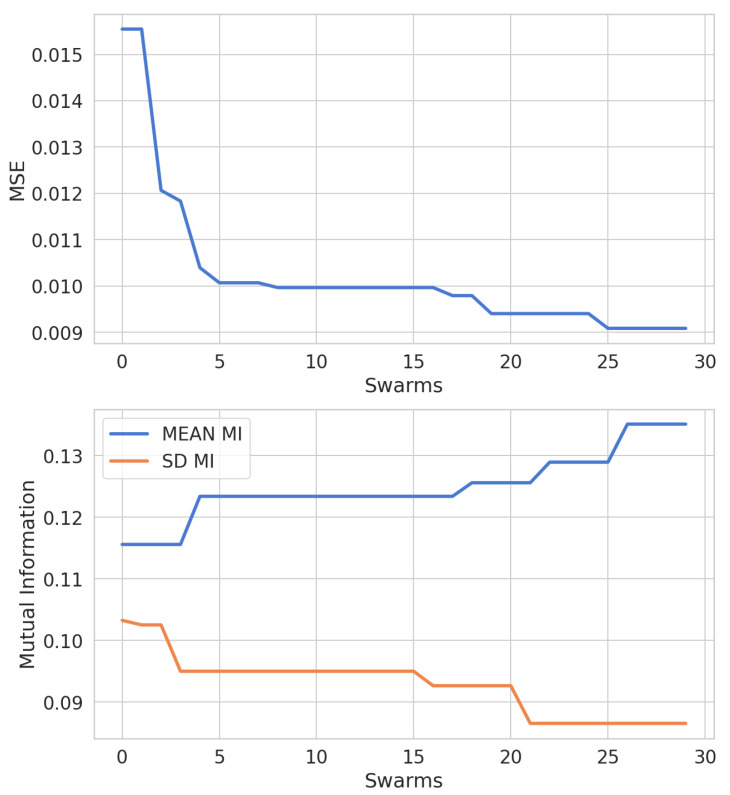
MSE, mean, and sd MI between input representations and output variables for the debutanizer column.

**Figure 6 sensors-22-06887-f006:**
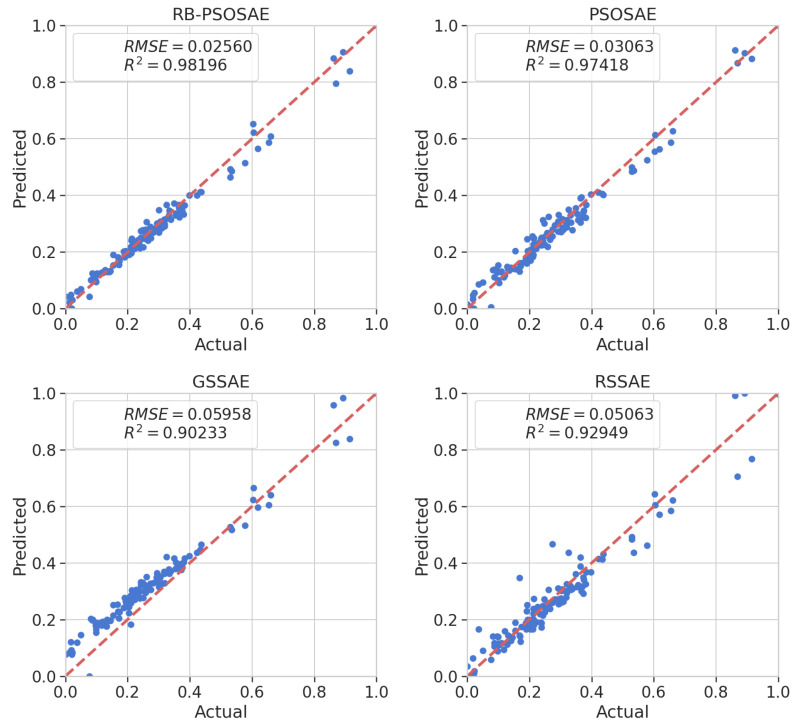
Real values of butane content and the predicted values using RB-PSOSAE, PSOSAE, GSSAE, and RSSAE search methods.

**Figure 7 sensors-22-06887-f007:**
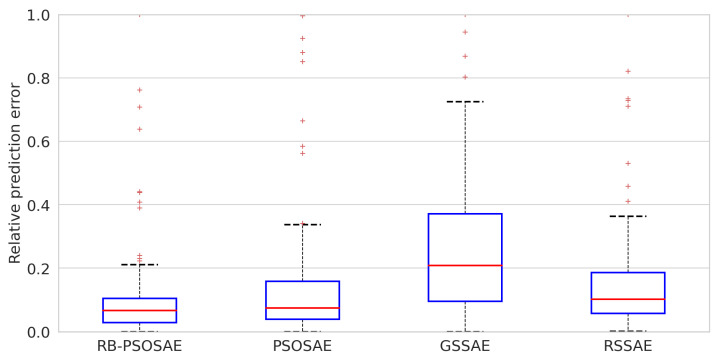
Relative prediction error of testing results for the debutanizer column process using RB-PSOSAE, PSOSAE, GSSAE, and RSSAE search methods.

**Table 1 sensors-22-06887-t001:** Description of debutanizer column process variables.

Variable	Variable Description	Unit
u1	Top temperature	°C
u2	Top pressure	kg/cm^2^
u3	Reflux flow	m^3^/h
u4	Flow to next process	m^3^/h
u5	Sixth tray temperature	°C
u6	Bottom temperature A	°C
u7	Bottom temperature B	°C
Output	Butane C4 content in IC5	-

**Table 2 sensors-22-06887-t002:** Prediction performance of debutanizer column soft-sensor models.

Model	RSME±SD	$R^{2}$
RSSAE	0.050626±0.0034	0.929491
GSSAE	0.056583±0.0041	0.902333
PSOSAE	0.030635±0.0023	0.974181
RB-PSOSAE	0.025604±0.0014	0.981965

## Data Availability

Original debutanizer column and SRU datasets are available http://www.springer.com/cda/content/document/cda_downloaddocument/9781846284793_material.zip?SGWID=0-0-45-349600-p168288081 (accessed on 3 January 2022).

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
