# Peer review of "Industrial Soft Sensor Optimized by Improved PSO: A Deep Representation-Learning Approach"

_sensors, 2022, doi:10.3390/s22186887_

Round 1

Reviewer 1 Report

The article reports on optimization of an industrial sensor with use of improved Particle Swarm Optimization. In fact, algorithms of Particle Swarm Optimization are gaining more and more popularity, and the correlated coefficients of determination derived by application to industrial processes are getting very close to 1. In the present study, the method was applied to an industrial debutanizer process showing considerable good performance of the algorithm. The main criticism could be that 3 of the 4 present authors have recently published very similar work cited as reference [13] in the present manuscript. Insofar, the novelty of the latest manuscript is limited. Nevertheless, it includes some new results so that the article could be published in this journal.

Minor comments:

i)                    In the title, the word “optimized” was misspelled.

ii)                  In Fig. 5, axis labels should be added to the diagrams.

Author Response

Response to Reviewer #1

Manuscript ID: sensors-1818129

Title: Industrial Soft Sensor Optimized by improved PSO: A Deep Representation-Learning Approach

Dear Reviewer #1,

First, the authors would like to thank you for your comments, suggestions, and the time and effort you have dedicated to providing valuable feedback about our manuscript. The authors are grateful for the insightful thoughts, which raise practical issues to improve the manuscript. We seriously considered all the comments to build the revised manuscript draft, and all the changes are in red. The revised text and explanations point-by-point are given below.

Reviewer #1 COMMENTS

The main criticism could be that 3 of the 4 present authors have recently published very similar work cited as reference [13] in the present manuscript.

Response: Many thanks for your comment. In fact, as in [13], we used the same case study, the debutanizer column, the SAE as a soft sensor strategy, and the mutual information index (MI). However, different from [13] the MI is applied as an improvement to the PSO algorithm, which has the responsibility of selecting the best SAE structure from the MI and MSE analysis, thus automatically performing the extraction of relevant features from the data used. Thus, our work presents an automatic hyperparameter selection strategy for deep neural networks, saving the designer the manual tuning work and enabling the selection of optimized parameters.

In the title, the word “optimized” was misspelled.

Response: Many thanks for your comment. We have corrected the title.

In Fig. 5, axis labels should be added to the diagrams

Response: Many thanks for your suggestion. We have inserted the information in the picture.

Reviewer 2 Report

Comment

This paper titled “Industrial Soft Sensor Optimizided by improved PSO: A Deep Representation-Learning Approach” proposed the RB-PSOSAE, a Representation-Based Particle Swarm Optimization with a modified evaluation function to optimize the hyperparameter set of a Stacked AutoEncoder-based soft sensor. Thus, the testing results demonstrate better prediction performance for debutanizer columns process parameters. So this article has the referential Value in a certain sense. However, the following concerns and doubts need to be addressed before the paper can be further considered.

(1)   Many papers on soft sensing methods of process parameters of debutanizer have been published , such as the extreme learning machineELM) modeling method based on the hybrid frog leaping algorithm, please compare their effectiveness.

(2)   Please explain the process time-delay characteristics and prediction of dynamic drift for primary variables at certain time lapse, some studies show that a deep Elman neural network model composed of multilevel Elman neural networks can realize real-time soft measurement of butane content.

(3)   Referring to the marked paper.

This is expectant that the manuscript would be further improved.

Author Response

Response to Reviewer #2

Manuscript ID: sensors-1818129

Title: Industrial Soft Sensor Optimized by improved PSO: A Deep Representation-Learning Approach

Dear Reviewer #2,

First, the authors would like to thank you for your comments, suggestions, and the time and effort you have dedicated to providing valuable feedback about our manuscript. The authors are grateful for the insightful thoughts, which raise practical issues to improve the manuscript. We seriously considered all the comments to build the revised manuscript draft, and all the changes are in red. The revised text and explanations point-by-point are given below.

Reviewer #2 COMMENTS

Many papers on soft sensing methods of process parameters of debutanizer have been published, such as the extreme learning machine (ELM) modeling method based on the hybrid frog leaping algorithm, please compare their effectiveness.

Response: Many thanks for your suggestion. As requested, we searched for papers that used ELM and the hybrid frog leaping algorithm to select the debutanizer column's hyperparameters on significant platforms such as SciELO, IEEE Explorer, Google Scholar, and others. However, due to the review time, we could not locate papers that applied both techniques for the evaluated case study. Nevertheless, we highlight that our work compares the proposed method to the traditional PSO, Random Search, and Grid Search algorithms regarding the selection of hyperparameters. If you can provide the reference, we will be very grateful and perform the necessary comparisons.

Please explain the process time-delay characteristics and prediction of dynamic drift for primary variables at certain time lapse, some studies show that a deep Elman neural network model composed of multilevel Elman neural networks can realize real-time soft measurement of butane content.

Response: Many thanks for your comment. In the case study of the debutanizer column, data was obtained with a sampling time of 6 minutes (Ts). An analyzer is typically used to obtain the butane content, although this adds a long delay estimated at 45 minutes. Similar processes can use chromatographs or laboratory analysis, where the time required to estimate the butane content is much longer. The model proposed in our work can estimate the butane content with 6 minutes delay, which, compared to the analyzer, is a much shorter time. Based on your comment, we have changed the article to improve the clarity of the information, as shown below:

“Table 1 lists the process variables present in the debutanizer column process. The study-case debutanizer offers 2,384 data samples for each process variable, with a sampling time Ts = 6 min.”

Referring to the marked paper.

Response: Many thanks for your comment. We have corrected the highlighted parts.
